# Phytochemical, Pharmacological, and Molecular Docking Study of Dry Extracts of *Matricaria discoidea* DC. with Analgesic and Soporific Activities

**DOI:** 10.3390/biom14030361

**Published:** 2024-03-18

**Authors:** Janne Sepp, Oleh Koshovyi, Valdas Jakštas, Vaidotas Žvikas, Iryna Botsula, Igor Kireyev, Hanna Severina, Oleksandr Kukhtenko, Kaisa Põhako-Palu, Karin Kogermann, Jyrki Heinämäki, Ain Raal

**Affiliations:** 1Institute of Pharmacy, Faculty of Medicine, University of Tartu, Nooruse 1, 50411 Tartu, Estonia; janne.sepp@ut.ee (J.S.); oleh.koshovyi@ut.ee (O.K.); kaisa.pohako@ut.ee (K.P.-P.); karin.kogermann@ut.ee (K.K.); jyrki.heinamaki@ut.ee (J.H.); 2Pharmacognosy department, National University of Pharmacy, 61002 Kharkiv, Ukraine; 3Institute of Pharmaceutical Technologies, Lithuanian University of Health Sciences, 44307 Kaunas, Lithuania; valdas.jakstas@lsmu.lt (V.J.); vaidotas.zvikas@lsmu.lt (V.Ž.); 4Department of Clinical pharmacology and clinical pharmacy, National University of Pharmacy, 61002 Kharkiv, Ukraine; botsula.iv@gmail.com (I.B.); ivkireev@ukr.net (I.K.); 5Pharmaceutical chemistry department, National University of Pharmacy, 61002 Kharkiv, Ukraine; severina.ai@ukr.net; 6Pharmaceutical Technology of Drugs Department, National University of Pharmacy, 61002 Kharkiv, Ukraine; kukhtenk@gmail.com

**Keywords:** pineapple weed, herb, extraction, terpenoids, polyphenols, cytotoxicity, analgesic activity, soporific effect

## Abstract

Pineapple weed *(Matricaria discoidea* DC.) is a widespread plant in Europe and North America. In ethnomedicine, it is well-known for its anti-inflammatory and spasmolytic activities. The aim of this research was to develop novel methods of *M. discoidea* processing to obtain essential oil and dry extracts and to investigate their phytochemical compositions. Moreover, the molecular docking of the main substances and the in vivo studies on their soporific and analgesic activities were conducted. The essential oil and two dry extracts from *M. discoidea* were prepared. A total of 16 phenolic compounds (seven flavonoids, seven hydroxycinnamic acids, and two phenolic acids) in the dry extracts were identified by means of UPLC-MS/MS. In the essential oil, nine main terpenoids were identified by gas chromatography (GC). It was shown that phenolic extraction from the herb was successful when using 70% ethanol in a triple extraction method and at a ratio of 1:14–1:16. The in vivo studies with rodents demonstrated the analgesic activity of the *M. discoidea* extracts and improvements in the sleep of animals. The dry extracts of *M. discoidea* did not show any toxicity. The molecular docking analysis showed a high probability of COX-1,2 inhibition and NMDA receptor antagonism by the extracts.

## 1. Introduction

Pineapple weed (*Matricaria discoidea* DC., syn. *Chamomilla suaveolens* (Pursh) Rydb., syn. *M. suaveolens* (Pursh) Buch., syn. *M. matricarioides* (Less.) Porter) is a species from the *Chamomilla* genus of the *Asteraceae* family. This plant is widely spread in Europe and North America, but it is not cultivable. *M. discoidea* has an ethnomedical background dating back to the 19th century, with the plant having been mainly used in the form of tea or a tincture for its anti-inflammatory and spasmolytic properties [1]. From a historical point of view, it is interesting to mention that the U.S.S.R. Pharmacopoeia [2] gave approval for the external use of *M. discoidea* as an additional raw material for the inflorescences of German chamomile (*M. chamomilla* L.).

In our previous study, a total of 44 compounds (essential oils) were found in *M. discoidea*. Biologically, the most relevant compounds were (Z)-enyne-dicycloether, (E)-β-farnesene, geranyl isovaleriate, palmitic acid, and myrcene [3]. In addition, it was shown that the quantitative content of the essential oils in the different aerial parts of *M. discoidea* does not vary significantly. Therefore, the use of herbs instead of inflorescences is more beneficial in terms of biomass [3,4]. Our findings on the total content of polyphenols, flavonoids, and coumarins supported this approach. It was also found that dicaffeoylquinic acids, chlorogenic acids, ferulic acid glycoside, quercetin galactoside, malonylapigenin glucoside, apigenin acetylglucoside, quercetin, luteolin, and apigenin glycosides are the main polyphenols in *M. discoidea* [4]. The compositions of *M. discoidea* and *M. chamomilla* have been reported in several publications [5,6].

Since *M. discoidea* and *M. chamomilla* have phytochemical compositions rich in bioactive substances, it is important to continue conducting in-depth studies with the purpose of improving the medicinal use of both plants. The development of new zero-waste technologies for isolating essential oils from these medicinal plants would be a sustainable and relevant approach at a global level since the supply of plant-origin materials is limited. Furthermore, such novel isolation technologies can make the production of plant-origin medicines more profitable, enable a more rational consumption of plant raw materials, and reduce the negative influence of pharmaceutical production on the environment [7,8]. To date, *M. discoidea* is widely used in folk medicines as a tincture or decoction, but the application of such preparations limits compliance from patients. In addition, the waste from the production of such tinctures and essential oils contains a significant amount of biologically active substances (BASs). Therefore, it is relevant to find novel methods and optimize the established methods for preparing medicines and food supplements of plant origin.

The aim of this research was to develop novel methods of *M. discoidea* processing to prepare essential oil and dry extracts and to study their phytochemical compositions. In addition, the affinity of the corresponding BASs to active the sites of the biotargets responsible for inducing the plant’s analgesic and soporific activities was predicted by molecular docking. The cytotoxicity of the BASs in vitro and the analgesic and soporific activities of the BASs in vivo were also investigated.

## 2. Materials and Methods

### 2.1. Plant Material and Reagents

The whole wild-growing *M. discoidea* (2.0 kg) aerial parts were collected during the plant’s flowering period in July 2020 at Lake Veskijärv, located in Nõo township, Nõo municipality, Tartu, Estonia [58°16′30″ N, 26°31′32″ E]. The plant material was immediately cleaned of impurities, and the herb was dried for one week at an ambient room temperature of 22 ± 2 °C in a well-ventilated room with 21% humidity. The dried herb was stored in paper packages at room temperature and protected from light (in a locker) until further studies. The raw material was identified using the Key to Higher Plants of Ukraine [9]. The voucher specimen #Ast/Mat/D13 of the plant is available in the Institute of Pharmacy, University of Tartu, Tartu, Estonia. The loss on drying (LOD) was determined using an MB23/MB25 Moisture Analyzer. Three parallel measurements showed a 6.5% loss on drying for *M. discoidea*.

The inflorescences of *M. chamomilla* were supplied by an Estonian herb farm, MK Loodusravi OÜ. The company operates according to a manufacturing activity license given by the Estonian State Agency of Medicines, which guarantees a proper manufacturing process and quality. The inflorescences of the cultivated plant were harvested in the summer of 2020 in the southern part of Estonia, Viljandi County, North Sakala municipality. The farm produces ecologically clean herbs.

*Reagents*. Deionized water was produced using a Millipore Simplicity UV station (Merck Millipore, Burlington, MA, USA). Acetonitrile, formic acid, and ethanol were purchased from VWR (Radnor, PA, USA). Chlorogenic acid, rutin, and gallic acid were purchased from Carl Roth (Karlsruhe, Germany). Aluminum chloride, hydrochloric acid, sodium nitrite, sodium molybdate, and sodium hydroxide were purchased from Sigma-Aldrich (Sant Louis, MI, USA). The chemical standards used for the HPLC analysis were previously isolated and identified at the Institute of Pharmaceutical Technologies, Lithuanian University of Health Sciences (Kaunas, Lithuania).

### 2.2. Isolation of Essential Oil

#### 2.2.1. Isolation of Essential Oil for Phytochemical and Pharmacological Analyses

The *M. discoidea* essential oil was obtained through distillation for 4 h, according to the Monograph “*Matricaria flower (Matricariae flos)*” in the European Pharmacopoeia [10]. For distillation, 300 mL of distilled water and 30 g of dried and cut plant material were placed in a 1000 mL round-bottom flask. To dissolve the obtained essential oil, 0.5 mL of cyclohexane was used. GC was performed on the same day after distillation.

#### 2.2.2. Isolation of Essential Oil for Cytotoxicity Studies

The essential oils of *M. chamomilla* flowers and *M. discoidea* were isolated from dried herbal samples. It was not possible to use the European Pharmacopoeia distillation method described in Monograph 10.8 “*Matricaria flower (Matricariae flos)”* since the essential oils were to be used in eukaryotic cell cytotoxicity experiments. There were two reasons for this: firstly, the solvent (cyclohexane) itself is toxic to cells, and secondly, the amount of plant material used in this method was too small to obtain a measurable volume of essential oil. Nevertheless, following the Ph. Eur. standards as much as possible, dried herbs were used, the distillation time was 4 h, and distilled water was used. Our preliminary experiments showed that without using an organic solvent, it was not possible to isolate a measurable amount of essential oil from the herbs. Therefore, it was decided to use a distillation system that uses only distilled water and larger amounts of herbs, allowing us to obtain a measurable amount of pure essential oil. A distillation extractor with a capacity of 12 L was used, along with 1 kg of dried herb and 5 L of distilled water. An induction stove was used for heating. The distillation time was 4 h and the distillation rate was regulated (by an induction stove), thus avoiding rapid temperature changes. To separate the pure essential oil from the aromatic water, a separation funnel was used (Cider Mill, n.d.). A total of 1 mL of *M. discoidea* essential oil and *M. chamomilla* essential oil each was obtained. The essential oils were stored in Eppendorf Tubes^®^ at −18 °C.

### 2.3. Preparation of Extracts

To isolate the essential oils, dried *M. discoidea* (100.0 g) was distilled in water *R* (1250.0 mL) for 3 h in an essential oil extractor (Albrigi Luigi SRL, Stallavena, Italy). The amount of essential oil in the dry herb was 3 mL/kg. The cooled aqueous distilled liquid was filtered from the chamomile herb. A total of 862 mL of the extract was obtained, and the dry residue accounted for 3.0 ± 0.3%. This extract was evaporated to a dry extract (P1) in a SCANVAC COOLSAFE 55-4 Pro (LaboGene ApS, Denmark) lyophilic apparatus. The yield of extract P1 was 25.9%.

Dried *M. discoidea* (500.0 g) was extracted by maceration with 2500 mL of 70% ethanol at room temperature (22 ± 2 °C) overnight. The same extraction was repeated five times with a new extractant (1000.0 mL each) to assess the rational extraction multiplicity. For preparing the final product, only the first three liquid extracts were used. These extracts were combined, and the mixture was allowed to sediment for a couple of days and then filtered. The combined liquid extract was further evaporated into a more concentrated extract with a Buchi B-300 rotary vacuum evaporator (Buchi AG, Flawil, Switzerland). Finally, the extract was reduced into a dry extract (P2) by drying in the lyophilic apparatus. The yield of extract P2 was 20.1%.

The remaining *M. discoidea* meal was extracted with water *R* (1000.0 mL) by boiling the mixture for 30 min and infusing it for 12 h. The liquid extract was filtered and evaporated in a lyophilic dryer apparatus to form a dry extract (P3). The yield of extract P3 was 8.4%.

For the cytotoxicity studies, the ethanol extracts were prepared as follows: both *M. chamomilla* and *M. discoidea* samples (5 g) were extracted with 125 mL of 96% (*w*/*v*) ethanol for 30 min at an ambient room temperature (22 ± 1 °C) using an MSO1 magnetic stirrer (ELMI, Riga, Latvia). For both herbs, the total amount of extract prepared was 125 mL. The extract was subsequently filtered and stored in dark glass bottles at 4 °C.

### 2.4. Phytochemical Analysis

#### 2.4.1. Gas-Chromatographic Analysis of Essential Oil

An Agilent GC 7890a gas chromatograph (GC) (Santa Clara, CA, USA) with a flame ionization detector and Agilent Open Lab CDS Chem Station software were used for the qualitative and quantitative analyses of the principal compounds (>1%) of the *M. discoidea* essential oil. Two fused silica capillary columns with the phases DB-5 and HP-Innowax (30 m × 0.25 mm, Agilent) were simultaneously used. Hydrogen was used as the carrier gas at a 3.0 mL/min flow rate and a split ratio of 1:150. The temperature program ranged from 50 to 250 °C at a rate of 2.92 °C/min, and the injector temperature was 250 °C.

The mean retention time and peak area of four parallel chromatograms were used for identifying *M. discoidea* essential oil principal compounds; their retention indices were compared, and their quantitative content was determined (%). The components were identified by comparing their non-polar DB-5 column retention indices to the corresponding values obtained from the databases and literature data [3,4,5,6,7].

#### 2.4.2. Measurement of Main Phenolics by Spectrophotometry

The measurement of total phenols, hydroxycinnamic acids, and flavonoids in the *M. discoidea* extracts was carried out using a Shimadzu UV-1800 spectrophotometer (Shimadzu Corporation, Japan).

Hydroxycinnamic acids (in terms of chlorogenic acid) were assayed after adding sodium molybdate, hydrochloric acid, and sodium nitrite at 525 nm [10,11].

A total of 0.250 g of the extract was dissolved in a 40% water solution of ethanol in a 25.0 mL volumetric flask, with constant stirring. Afterward, 1.0 mL of the solution was placed in a 10 mL volumetric flask, to which, the following were successively added: 2 mL of a 0.5 M solution of hydrochloric acid; 2 mL of a freshly prepared solution of 10 g of sodium nitrite *R* and 10 g of sodium molybdate *R* in 100.0 mL of water *R*; and 2 mL of diluted sodium hydroxide solution *R*. The final volume of the solution was adjusted to the mark with water *R*.

The compensation solution consisted of 1.0 mL of the extract solution mixed with 2 mL of a 0.5 M solution of hydrochloric acid and 2 mL of a diluted sodium hydroxide solution *R* in a 10 mL volumetric flask; the final volume of the solution was brought up to the mark with water *R*. The optical density of the test solution in a cuvette with a thickness of 10 mm was immediately measured at a wavelength of 525 nm, using the compensation solution as a comparison solution.

The amount of hydroxycinnamic acids, in terms of chlorogenic acid, in percent, was calculated according to the following formula:$$X=\frac{A*1000}{188*m},$$
where *A* is the optical density of the tested solution at a wavelength of 525 nm, and m is the weight of the tested extract (g).

The specific absorption index of chlorogenic acid was used, which is equal to 188.

Flavonoids were determined in terms of rutin after forming a complex with aluminum chloride at 417 nm [10,12].

A total of 0.250 g of the extract was dissolved in 70% ethanol in a 25.0 mL volumetric flask. A 2.0 mL volume of the solution was placed into a 25 mL volumetric flask, and 2.0 mL of 3% aluminum chloride in 96% ethanol was added; the final volume was brought up to the mark with 70% alcohol. After 30 min, the solution was filtered through a paper filter, the first portions of the filtrate were discarded, and the optical density of the obtained complex was measured using a spectrophotometer at a wavelength of 417 nm in a cuvette with a thickness of 10 mm. The reference solution was a solution containing 2.0 mL of the extract solution, adjusted up to the mark with 70% ethanol in a 25.0 mL volumetric flask.

In parallel, under the same conditions, an experiment using a standard solution of rutin was conducted. A 1.0 mL volume of a 3% alcohol solution of aluminum chloride was added to 1.0 mL of standard solution and diluted to 25.0 mL with 70% ethanol. As a comparison solution, a standard solution of rutin was added into a 25.0 mL volumetric flask and adjusted up to the mark with 70% ethanol.

The amount of flavonoids in the samples in terms of rutin was calculated as a percentage using the following formula:$$X=\frac{A_{1}*a_{0}*25*1*25*100*100}{A_{0}*a_{1}*25*2*25*\left(100-w\right)},$$
where *A*_1_ is the optical density of the test solution, *A*_0_ is the optical density of the standard rutin solution in complex with aluminum chloride, *a*_1_ is the weight of the extract (g), *a*_0_ is the weight of the rutin standard (g), and w is the mass lost during drying (%).

Preparation of rutin standard solution. A total of 0.01 g of rutin (FS 42-2508-87), dried at a temperature of 135 °C to a constant mass, was placed into a 25.0 mL volumetric flask, dissolved in 96% ethanol, and the volume of the solution was adjusted to the mark and stirred.

The amount of total phenolics was assayed in terms of gallic acid at 270 nm [11].

A total of 0.250 g of the extract was dissolved in a 40% water solution of ethanol in a 25.0 mL volumetric flask, with constant stirring. A 1.0 mL volume of the extract solution was placed into a 25.0 mL volumetric flask, and the volume was adjusted up to the mark with 40% ethanol. A 1.0 mL volume of this solution was diluted with the same solvent in a 10.0 mL volumetric flask. After that, the optical density of the solution was measured using a spectrophotometer at a wavelength of 270 nm in a cuvette with a thickness of 10 mm. A 40% aqueous ethanol solution was used as a comparison solution.

The amount of total phenolics (X) in the extract in terms of gallic acid was calculated as a percentage according to the following formula:$$X=\frac{A*25*25*10*100}{\begin{array}{c} 540*m*1*1*\left(100-w\right) \end{array}}$$
where *A* is the optical density of the test solution, *m* is the weight of the extract (g), 540 is the specific absorption coefficient of a solution of gallic acid in 40% alcohol at a wavelength of 270 nm, and *w* is the mass lost during the drying of the raw materials (%).

#### 2.4.3. Analysis of Phenolic Compounds by UPLC-MS/MS

Quantitative and qualitative analyses of the phenolics in the *M. discoidea* extracts was carried out using a UPLC-MS/MS system. Chromatographic separation was conducted with an Acquity H-class UPLC chromatograph (Waters, USA) equipped with a YMC Triart C18 column (100 mm × 2.0 mm; 1.9 µm). The column was maintained at constant temperature of 40 °C. The mobile phase was supplied at a flow rate of 0.5 mL/min. An aqueous formic acid solution (0.1%) was used as solvent A, and pure acetonitrile was used as solvent B. Gradient elution was applied under the following conditions: Solvent B, from 0 to 1 min at 5%; increase to 30% solvent B, from 1 to 5 min; linear increase to 50%, from 5 to 7 min; column wash with solvent B, from 7.5 to 8 min; and equilibrate column to initial conditions of 5% solvent B, from 8.1 to 10 min. A triple quadrupole tandem mass spectrometer (Xevo, Waters, USA) was used for the chemical structure analysis of the phenolic compounds. Negative electrospray ionization (ESI) was applied to generate ions for the MS/MS data acquisition. The MS/MS analysis settings were as follows: capillary voltage of negative 2 kV, the nitrogen gas for desolvation was heated to 400 °C, flow rate of 700 L/h, gas flow of 20 L/h, and a temperature of ion source at 150 °C. The qualitative determination of phenolics was performed by comparing their MS/MS spectral data and retention times with those of analytical grade standards. Linear regression fit models and the standard dilution method were used for the quantitative analysis of phenolics [11,13].

### 2.5. Molecular Docking of M. Discoidea BAS

The molecular docking experiments were carried out using AutoDock Vina and AutoDockTools 1.5.6 [14]. A macromolecule from the Protein Data Bank [PDB] was used as a biotarget: •PDB ID 6NCF, 3N8Y, 3LN1, 7EU7, 6X3W. The construction of a virtual database of candidate structures was performed using the BIOVIADraw 2021 program and saved in mol format. The structures were optimized by Chem3D with the MM2 molecular mechanics algorithm, saved in the.pdb format, and converted to .pdbqt using AutoDockTools-1.5.6. Discovery Studio Visualizer 2021 was adopted to remove the native protein ligands and solvents. The macromolecule was saved in the.pdb format. In AutoDockTools-1.5.6, polar hydrogen atoms were added to the protein structure, which was converted to the .pdbqt format. The size of the grid box and its center were determined by the native ligands:

LOX-5 (PDB ID 6NCF): x = 11.6, y = −23.38, z = −18.01; size x = 30, y = 28, z = 26;

COX-1 (PDB ID—3N8Y): x = 33.14, y = −44.49, z = −3.76; size x = 24, y = 22, z = 20;

COX-2 (PDB ID 3LN1): x = 18.84, y = −52.89, z = −53.81; size x = 22, y = 24, z = 24;

NMDAR (PDB ID 7EU7): x = 124.19, y = 125.67, z = 78.60; size x = 12, y = 14, z = 14;

GABAA (PDB ID 6X3W): x = 109.83, y = 93.68, z = 105.43; size x = 22, y = 18, z = 16.

AutoDock Vina was used for docking. The analysis and visualization of the docking results were performed using Discovery Studio Visualizer 2021 Client.

Macromolecules from the Protein Data Bank [15] were used as target proteins as follows:The lipoxygenase-5 (LOX-5) (PDB ID 6NCF) enzyme with a natural non-competitive inhibitor, pentacyclic triterpenoid acid (3α,8α,17α,18α-3-acetyloxy-11-oxours-12-en-23-oic acid; AKBA), in the active site [16];Cyclooxygenase-1 (COX-1) (PDB ID—3N8Y) [17] and cyclooxygenase-2 (COX-2) (PDB ID—3LN1) [18] enzymes in conformation with diclofenac and celecoxib, respectively;Ionotropic glutamate NMDA receptors in conformation with a non-competitive antagonist with direct actions: ketamine (7EU7) [19];The GABA receptor in conformation with the agonist phenobarbital (6X3W) [20].

### 2.6. Pharmacological Study

Adult male random-bred albino mice were used for investigating the analgesic activity of the essential oils and extracts. Adult 12–18-month-old male out-bred albino rats were used in the soporific activity study. The mice and rats were housed in standard polypropylene boxes in a well-ventilated room with 50% relative humidity at 22 ± 2 °C, with a 12 h light/dark cycle and free access to water and food in the vivarium of the National University of Pharmacy (Kharkiv, Ukraine).

All pharmacological studies were performed according to the rules of the “European Convention for the Protection of Vertebrate Animals Used for Experimental and Other Scientific Purposes” (Strasbourg, 1986), Directive 2010/63/EU of the European Parliament, and the Council of the European Union (2010) on the protection of animals used for scientific purposes. The Order of the Ministry of Health of Ukraine No. 944, “On Approval of the Procedure for Preclinical Study of Medicinal Products and Examination of Materials for Preclinical Study of Medicinal Products” (2009), and the Law of Ukraine No. 3447-IV, “On the protection of animals from cruel treatment” (2006), were also strictly followed. The present research was approved by the Bioethics Commission of the National University of Pharmacy (protocol No. 4 from 03.10.2023) [21,22,23,24,25].

#### 2.6.1. Cytotoxicity Studies

The cytotoxicity studies were performed on two cell lines: baby hamster kidney fibroblast (BHK-21) and human bone osteosarcoma epithelial (U2OS) cells. U2OS cells were grown in DMEM (Dulbecco’s Modified Eagle Medium, Gibco) supplemented with 10% FBS (fetal bovine serum, Gibco), 100 μg/mL of penicillin, and 100 μg/mL of streptomycin. BHK-21 cells were grown in GMEM (Glasgow Minimum Essential Medium, Gibco) supplemented with 10% FBS, 2% TPB (tryptose phosphate broth, Gibco), 2% 1M (238.3 mg/mL) HEPES buffer (Mediatech, Inc., Manassas VA, USA) solution, 100 μg/mL of penicillin, and 100 μg/mL of streptomycin. Both media contained phenol red and 4.5g/L glucose. The cells were maintained at 37 °C in a 5% CO_2_ incubator. The cells were monitored and visualized using an optical microscope (Primovert inverted microscope).

A Countess^TM^ automated cell counter (Invitrogen^TM^, Carlsbad, CA, USA) was used for counting the cells using a 0.4% trypan blue stain (Invitrogen^TM^). The U2OS cells were counted and documented. The number of BHK-21 cells per well (10,000 cells per well) was fixed at the beginning of the experiment.

The following four samples were investigated in the cytotoxicity studies: *M. chamomilla* essential oil, *M. discoidea* essential oil, *M. chamomilla* ethanol extract, and *M. discoidea* ethanol extract. For the extracts, 96% ethanol and pure growth medium were used as controls. For the essential oils, pure growth medium was used as the control. A series of 2-fold dilutions were prepared for all samples as well as ethanol as a control in 1.5 mL microtubes. For each dilution, three replicates were used. The experiments were repeated at least in triplicate.

A 100 µL volume of the sample was added to a 96-well plate and incubated for 48 h at 37 °C in a 5% CO_2_ incubator. The maximal non-cytotoxic concentrations (dilutions) of the extracts were determined using a tetrazolium dye MTS Proliferation Assay (Biovision). The MTS reagent (10 µL per well) was added to the wells and incubated for 1 h at 37 °C in a 5% CO_2_ incubator. After incubation, the absorbance was measured at 490 nm using a microplate reader (Tecan Sunrise). The analysis was conducted in a biosafety level 2 (BSL-2) laboratory using aseptic technique principles. All solvents used were of pharmaceutical grade.

#### 2.6.2. Analgesic Activity

The analgesic activity of the *M. discoidea* herb extracts (P1, P2, P3), as well as that of acetaminophen (Paracetamol-Zdorovye capsules, 500 mg; Pharmaceutical company «Zdorovye», Kharkiv, Ukraine), were investigated in mice by using an established hot plate test [26].

The test was performed once, and no washing period was used. The period of acclimatization and quarantine was 14 days. The animals (22–40 g in weight) were randomly assigned into 11 groups, with 6 mice per group.

Group 1—The control group that received a 0.9% solution of NaCl at a dose of 1 mL per 100 g of body weight;

Groups 2, 3, 4—The mice received 25, 50, and 100 mg/kg of the P1 extract;

Groups 5, 6, 7—The mice received 25, 50, and 100 mg/kg of the P2 extract;

Groups 8, 9, 10—The mice received 25, 50, and 100 mg/kg of the P3 extract;

Group 11—The positive control group that received 50 mg/kg of acetaminophen.

Before the test, the rodents were not fed for 2 h. The *M. discoidea* extracts were administered intragastrically as an aqueous suspension 30 min before the animals were placed on the hot plate device. Aqueous suspensions of ethanol extracts were prepared fresh. The protocol is detailed below.

After the consumption of the extracts or the reference drug, the mouse was placed on a thermostatic hot plate (55 °C) for 30 min. The reaction latency period was fixed as the time needed for the mouse’s response to the heat stimulation by licking the paw, flinching, or jumping [27,28]. The mice were observed for 0.5, 1, 2, 3, and 4 h after the extract or control drug administration. The analgesic activity criterion was an increase in the latency period after administering the extracts compared to the control. To avoid thermal damage to the tissues of paw, the hot plate exposure time for the mice did not exceed 60 s. The extracts’ analgesic effect was calculated according to the equation:
AA = (T_e_ − T_c_)/T_c_ × 100%;
where
AA = the analgesic activity (%);T_e_ = the difference in the corresponding response latency period in the group of animals after administering the extracts;T_c_ = the difference in the response latency period in the control group after administering the solvent.

#### 2.6.3. Soporific Activity

The soporific effects of the three extracts (P1, P2, and P3) and the reference drugs sodium thiopental lyophilizate (PLC “Kiivmedpreparat”, Kyiv, Ukraine) and “Valerian syrup AN NATUREL” syrup (LLC Beauty and Health, Kharkiv, Ukraine) were studied using a sodium thiopental-induced sleeping time test [29,30].

The white male rats (190–280 g) were divided into 11 groups (6 animals per group):

Group 1—The control group that received a 0.9% solution of NaCl at a dose of 1 mL per 100 g of body weight;

Groups 2, 3, and 4—The rats received 25, 50, and 100 mg/kg of the P1 extract;

Groups 5, 6, and 7—The rats received 25, 50, and 100 mg/kg of the P2 extract;

Groups 8, 9, and 10—The rats received 25, 50, and 100 mg/kg of the P3 extract;

Group 11—The Valerian group that received 2.14 mg/kg of Valerian syrup.

The sleep duration was recorded as the time period during which the rats were in a lateral position.

### 2.7. Statistical Analysis

The “Statistical Analysis of the Results of a Chemical Experiment” monograph of the European Pharmacopoeia was used to calculate the mean and standard deviation (SD) [10]. For the phytochemical content analysis, a minimum of three measurements were conducted. The data are presented as mean values ± SD [10]. For the pharmacological study, the results are presented as the average of six measurements ± SD using the Student’s *t*-test. A *p* < 0.05 was considered statistically significant. To analyze the cytotoxicity results, confidence intervals and the z-test were used.

## 3. Results

The three dry extracts (P1, P2, and P3) prepared from *M. discoidea* were hygroscopic brown powders with a characteristic smell. The P2 extract had a greenish tint, and it became a viscous, thick mass during storage. The loss of drying values for the extracts ranged from 4.1% to 4.8% [10].

### 3.1. Phytochemical Composition of Essential Oil and Dry Extracts

A total of nine main terpenoids were identified and quantified, which comprised approximately 96% of the *M. discoidea* essential oil composition (Table 1).

The main phenolics of the *M. discoidea* dry extracts were identified by UPLC-MS/MS (Table 2). The contents of phenolic compounds, hydrocinnamic acids, and flavonoids were also determined using established Pharmacopoeia spectrophotometric methods (Table 2, Figure 1).

A total of 16 phenolic compounds (7 flavonoids, 7 hydroxycinnamic acids, and 2 phenolic acids) were identified in the *M. discoidea* dry extracts.

### 3.2. Optimization of a Dry Extract P2 Preparation

To determine the effective ratio of the extractant (70% aqueous ethanol solution) to *M. discoidea*, the effects of the extractant ratio to raw materials (DIR), and the extraction multiplicity on the yield of the extractive substances and the BAS content (phenolics, flavonoids, and hydroxycinnamic acids) were investigated. For such experiments, 500.0 g of *M. discoidea* was used and six sequential stages of extraction were conducted at room temperature (22 ± 2 °C) under normal air pressure using a laboratory percolator. The coefficient of *M. discoidea* herb for the absorption of a 70% aqueous ethanol solution was 1.96. The amount of phenolics, flavonoids, and hydroxycinnamic acids in the liquid extracts was measured using established Pharmacopoeia methods (Table 3).

Using these results (Table 3), the optimization of the extraction process was carried out using each stage’s mass yield coefficient (m_i BAS_/V_i extractant_) of the different compounds [32,33]. The diagram of these compounds’ dependence on the extraction rate is shown in Figure 2.

The impacts of the DIR and extraction multiplicity on the yield of extractive substances were investigated. The polynomial equations describing the correlation between the BAS yields and the extractant-to-raw material ratio of were used to optimize the extraction process.

### 3.3. In Silico Prediction of the Pharmacological Activity of M. discoidea BAS

The docking of native reference ligands into the corresponding active sites (i.e., a re-docking procedure) was used to evaluate the efficiency of the methodologies and docking parameters in reproducing our experimental conformational placement data.

The reproducibility of all connections during fixation in the active sites (from the literature and established in our study) was successfully realized. The values of the root-mean-square deviation (RMSD) between the native and reference conformations were calculated using the ProFit Results online resource, which were 2.023 Å (AKBA) [34], 1.001 Å (diclofenac), 1.952 Å (celecoxib), 1.998 Å (TK-40), 2.043 Å (ketamine), and 2.123 Å (phenobarbital). These results confirmed the reproducibility of the experimental data and the validity of method.

The level of affinity to the corresponding active site of the biotarget was assessed based on the binding energy parameter (kcal/mol) relative to the reference ligands (Table 4).

The types of interactions between the ligands and the amino acids of the active site were analyzed, and the conformational arrangement relative to the reference ligand was determined for the substances with the highest binding energy (Table 5).

### 3.4. Pharmacological Study

#### 3.4.1. Cytotoxicity Study

The cytotoxicity study enabled us to determine the dilutions for both the chamomile alcohol extracts and pure essential oils that are non-toxic to eukaryotic cells. A non-toxic concentration is defined as the concentration of a sample at which 80% of the cells exposed to the sample remain alive in an MTS assay. Preliminary studies were conducted, which enabled us to determine the suitable dilution ranges for the extracts and essential oils. Figure 3 shows that the cytotoxicity of the chamomile essential oils occurred at much lower concentrations (dilutions of 1:800 to 1:1000) compared to the alcohol extracts (dilutions of 1:20 to 1:40). There were no significant differences observed in the cytotoxic concentration values between the two different cell lines (U2OS, BHK-21) used. In addition, both chamomiles (M. discoidea and M. chamomilla) had similar toxicity concentration profiles.

#### 3.4.2. Analgesic Activity

The hot plate test was used to study the *M. discoidea* dry extracts’ effect on the nociceptive system in mice; the results are summarized in Table 6.

The administration of the dry extracts prolonged the reaction time to the thermal irritant. However, the analgesic activity of the P2 dry extract (at all three doses studied) was higher in comparison to the mice in the control group that received acetaminophen.

#### 3.4.3. Soporific Activity

Herbal extracts have been a valuable source of new therapeutics for the treatment of various diseases or disorders; for example, insomnia. Chamomile tea and chamomile essential oils are widely used for their sedative–hypnotic effects [35,36]. In this study, the soporific effect of the *M. discoidea* dry extracts was assessed by measuring their effects on sodium thiopental-induced sleep. The duration of sleep was determined, and the results are summarized in Table 7.

The consumption of the *M. discoidea* extracts (20 min before thiopental sodium administration) extended the sleep duration. The group of rats treated with the P2 extract (100 mg/kg) exhibited a higher sedative effect; thus, this extract prolonged the sleeping time in rodents by 2.8 times compared to the control group.

## 4. Discussion

The European Pharmacopoeia (Ph. Eur.) [10] has a specific monograph for *M. chamomilla* flowers and chamomile essential oil, but there is no monograph for the herbal drug or oil of *M. discoidea*. This is obviously due to the fact that there are relatively few studies on the materials from this species. Unlike *M. chamomilla* essential oil, the composition of *M. discoidea* oil has only been investigated in few publications, including in Estonia [1,4,6,37,38] and in other countries [39,40,41,42,43]. *(E)-*ß-farnesene, (Z)-enyne-dicycloether, and geranyl isovalerate were reported as the main compounds of *M. discoidea* essential oil in the abovementioned studies. Thus, the essential oil of *M. discoidea* showed a content of terpenoids corresponding to previous results.

The studies found nine main compounds in the essential oil (which was hydrodistilled from *M. discoidea*), which are listed in Table 1. The same terpenoids were also found in high concentrations in the oil of *M. discoidea* flowers [5]. The main terpenes of the *M. discoidea* essential oil are *(E)*-farnesene (42.5%), geranyl isovalerate (29.5%), *(Z)*-enyne-dicycloether (8.9%), and myrcene (8.0%) (Table 1). The contents of these compounds in the flowers and different aerial parts of *M. discoidea* were between 19.5 and 30.2%, 8.4 and 15.1%, and 17 and 40.7%, respectively [4].

The essential oil of *M. discoidea* can be used as an additional source of chamomile oil but the content is different compared to that of *M. chamomilla* flowers. Both Estonian [1,3,4] and former Soviet Union scientists [41] have come to the same conclusion: since the essential oils’ chemical composition is similar, the herb of *M. discoidea* can be a substitute for *M. chamomilla* flowers. On the other hand, it is worth mentioning that this conclusion was not based on the chemical composition only, since the most important factor is the effect of the two chamomile species on the human body. In this study, it was found that the biological activities of the two studied chamomile extracts were not equivalent. Moreover, the chemical compositions of the two plant species are different since the main terpenes in the *M. chamomilla* flower essential oil are bisabolol and its oxides, while the essential oil of *M. discoidea* does not contain chamazulene [4], and the oil has low levels of bisabolol and bisabolol oxides.

The phenolic compositions of *M. discoidea* and *M. chamomilla* extracts are quite similar [4,9]. The dry extract of *M. discoidea* obtained using 70% ethanol contained less luteolin-4-O-glucoside and isorhamnetin-3-glucoside but more 3,4-dihydroxyphenylacetatic acid and luteolin-7-O-glucoside than the *M. chamomilla* extract. The potential impact of phenolics on the pharmacological activity of chamomile is discussed below based on the molecular docking results of the *M. discoidea* BASs.

According to the results shown in Figure 2, it can be concluded that the effective extractant-to-raw-material ratio is 1:16–1:18. The yield of extractive substances reached a “plateau” at these concentrations and did not significantly increase with higher proportions of the extractant.

To extract flavonoids and hydroxycinnamic acids, an extractant-to-raw-material ratio of 1:12–1:14 is recommended. The yield of these active molecules did not significantly increase with higher proportions of the extractant, and it may increase the ballast substance output. Phenolics are more optimally extracted at a DER of 1:14–1:16. The application of these ratios in the extraction process enables us to achieve the highest yield of phenolic compounds.

The results presented in Figure 3 show that *M. discoidea* and *M. chamomilla* have similar cytotoxic profiles in different cell lines (U2OS, BHK-21). Cytotoxic effects from the essential oils occurred at much lower concentrations (dilutions of 1:800 to 1:1000) compared to the alcohol extracts (dilutions of 1:20 to 1:40). These results are important in the development of new herbal medicines. Based on the results obtained in the prediction of the pharmacological activity of BASs in *M. discoidea* extracts, the following conclusions were drawn. The highest degree of affinity to the LOX-5 inhibitor site was calculated for all types of luteolin glycosides (binding energy > −9 kcal/mol), and also for 3,5-dicaffeoylquinic acid (−9.0 kcal/mol) and rutin (−8.9 kcal/mol). These values were obtained even though the value of the scoring function was somewhat inferior to the reference ligand (−10.0 kcal/mol). The details of the interaction between these BASs and peptide residues of the target and their conformational arrangement relative to native AKBA are shown in Table 5 (as the example of luteolin-3,7-diglucoside and rutin).

For luteolin-3,7-diglucoside, it was possible to firmly fix all fragments of the molecule with both hydrophobic and hydrogen bonds (Figure 4). The compatible conformation with AKBA demonstrates the similarity of their fixation with the possibility of a sufficiently deep immersion into a hydrophobic cavity. On the other hand, rutin is unable to completely immerse itself in the binding pocket (in particular, in a chromene cycle), thus making the stable and strong fixation of rutin impossible. According to the results of the molecular docking analysis, luteolin and its glycosides, three dicaffeoylquinic acids, and hyperoside have the highest probability of being LOX-5 inhibitors.

Based on the affinity score for the active site of COX-1, most of the studied ligands were inferior to the reference ligand diclofenac (Table 4). It should be noted that there was a significant difference in the affinity of isorhamnetin (−7.8 kcal/mol) and its glycoside (−1.8 kcal/mol) to COX-1, which could be explained by the location in the active site and the inability of the glycoside to penetrate the hydrophobic pocket (Figure 5a). Instead, all fragments of isorhamnetin were fixed by a network of hydrophobic bonds. In particular, a benzopyranone ring formed a bidentate bond with Gly526 and a tetrahedral bond with Ala527 (Figure 5b). These amino acids are also involved in the fixation of diclofenac [17]. A similar conformational arrangement in space was also observed with luteolin (Table 5). The analysis of the molecular docking results shows that luteolin, isorhamnetin, and neo- and chlorogenic acids may have COX-1 inhibition activity.

The molecular docking in COX-2 revealed that all the studied ligands were inferior in terms of the degree of affinity compared to the reference drug celecoxib (Table 4). However, the indicators were rather low, and successful placement in the active site required the participation of backbones for the inhibitory ability [18] for luteolin, isorhamnetin, and 4,5-dicaffeoylquinic acid. The compatible arrangement of celecoxib in the active site suggests the requirement for the spatial positioning of substances with a benzopyranone fragment in their structure (Figure 6a). The nature of interactions with amino acid residues indicates the conformation stability (Figure 6a), and the overlapping of all parts of isorhamnetin and luteolin indicates a high probability of isorhamnetin and luteolin having inhibitory activity against COX-2. A high probability for COX-2 inhibitory activity was also predicted for 4,5-dicaffeoylquinic acid since it was completely and deeply immersed in the active site, thus occupying a spatial position almost identical to the native ligand (Figure 6b).

For most of the ligands studied here, the calculated values for the scoring functions for binding to the active site of the NMDA receptor were lower than that for the reference ligand ketamine (excluding vanillic acid, caffeic acid, and 3,4-dihydroxyphenylacetic acid). The inhibition of the NMDA receptor by ketamine occurs by fixing it in the channel pore through the participation of three hydrophobic bonds with Val644 and Leu642. Therefore, the amino acid interactions of the ligands were investigated, and the results showed that only dicaffeoylquinic acids and luteolin have the ability to bind the channel pore through interactions with Val644 and Leu642 (Table 5, Figure 7a). As shown in Figure 7a, a hydrophobic fixation is possible only through a benzopyranone ring interaction with Val644 (for example, in luteolin-7-O-glucoside). The other fragments of the molecule only formed hydrogen bonds with amino acids that are not essential for the manifestation of activity, and this does not ensure conformational stability and inhibition of the receptor.

An agonistic effect of the studied ligands on the GABA receptor through the active site bound by barbiturates is unlikely since the binding energy of the ligands was higher than that of the reference ligand phenobarbital (−7.6 kcal/mol). Moreover, the analysis of amino acid interactions showed the absence of hydrophobic bonds with the fixation of all fragments of the phenobarbital molecule with the amino acids of the active site (Ala291, Pro228, Leu231, Tyr294) [20]

The verification of the safety profile of chamomile extracts and oils is important before studying the therapeutic properties of these plants.

The results of the pharmacological study showed that the administration of *M. discoidea* extracts had an analgesic effect in mice by prolonging the latent time of discomfort in a hot plate test. The longest time period spent on the test plate was observed in the mice given 50 mg/kg of the P1 extract (two hours after administration, the time spent on the hot plate increased by 72%) and at 100 mg/kg (at one hour after administration, the time spent on the hot plate increased by 72%) in comparison to the control group (*p* < 0.05). The P3 extract increased the latency time by 47% (at one hour after administration; *p* > 0.05) and 57% (*p* < 0.05) at doses of 50 and 100 mg/kg, respectively.

The consumption of the P2 extract at all the doses showed significant (*p* < 0.05) analgesic activity compared to the administration of acetaminophen and the control group at different time points. The highest time spent on the hot plate was observed in the mice who received 50 mg/kg of the extract (at 120 min after administration) and 100 mg/kg (at 60 min after consumption). The time of discomfort occurrence in these groups was 81% and 82% higher compared to that of the control rats. The analgesic activity of these extracts (at all doses studied) did not demonstrate any significant difference in comparison with the analgesic effect from acetaminophen (*p* > 0.05).

The results from the hot plate test suggest that the *M. discoidea* extracts studied here have central analgesic effects. To date, no research on the analgesic activity of *M. discoidea* extracts has been published in the scientific literature. Recently, Chaves and co-authors investigated the nociceptive effect of dry *M. chamomilla* flowers using the formalin test [44]. The administration of 30 mg/kg of dry *M. chamomilla* flowers reduced nociception by 81% in Phase I and 96% in Phase II compared to a control group (10 mL/kg of saline solution). It is known that luteolin (the flavonoid obtained from the flavones of chamomile origin) has anti-inflammatory activity and can be used for the treatment of diseases accompanied by inflammation. Fan et al. (2018) studied the antinociceptive properties of luteolin in mice and found that luteolin has a significant and dose-dependent antinociceptive activity [45]. In the mouse experiments, luteolin prolonged the latency period in the hot plate test and inhibited the nociceptive responses in the formalin test.

Chamomile tea and essential oils have been widely used for their calming effects [33,35]. In the present study, we used a sodium thiopental-induced sleeping time test to investigate the sedative–hypnotic effects of the oils and extracts in rats. The sedative effect of the *M. discoidea* extracts (P1-P3) was confirmed, as they lengthened the sleeping time by 78.8–184.7%. The average sleep duration in the group that received 25 mg/kg of the P1 extract was 180.2 ± 11.4 min. The average sleep duration of the rats treated with 50 mg/kg and 100 mg/kg of the P1 extract was 171.7 ± 2.9 min and 170.0 ± 9.3 min, respectively. These sleep durations obtained with the P1 extract were significantly higher than that of the control group (by 106.3%, 96.6%, and 95.8%, respectively; *p* < 0.05). The sleep period was extended in the animal groups treated with the P3 extract at the abovementioned three doses by 89.7%, 78.8%, and 91.3%, respectively.

The P2 extract showed the best sedative activity at a dose of 100 mg/kg in rats, and this treatment increased the sleep period by 2.8 times compared to the control animal group. Additionally, the sleep duration was longer than that of the group treated with Valerian syrup (2.15 mg/kg). The sedative activity in the animal groups that were administered 25 mg/kg and 50 mg/kg of the extract were 178.2% and 146.8% higher than that observed in the control rats (*p* < 0.05). Therefore, the *M. discoidea* extracts show synergistic sedative and soporific effects with sodium thiopental.

Traditionally, chamomile is used as a sleep-inducer and mild tranquilizer. According to the literature, the sedative effect observed here could be related to apigenin, which binds to GABA and benzodiazepine receptors in the brain [35,46]. Apigenin also decreases cortisol plasma concentrations [47]. Shinomiya et al. studied the hypnotic activity of a chamomile extract using a sleep-disturbed rat model, and the authors reported a significant reduction in sleep latency with 300 mg/kg of the chamomile extract [48]. Amsterdam et al. (2009) reported that administration of standardized *M. recutita* (220–1100 mg of titrated extract depending on the response) resulted in a significant reduction in the anxiety scores of the Hamilton Anxiety Rating Scale compared to a placebo at the end of eight weeks of treatment [49]. The authors suggested that chamomile extracts may have anxiolytic effects in persons with mild to moderate generalized anxiety disorder [49].

## 5. Conclusions

In the present research, novel methods of *M. discoidea* processing for the isolation of essential oils and preparation of dry extracts were developed. A total of 16 phenolic compounds were identified in the *M. discoidea* herb dry extracts, and nine terpenoids were identified in the *M. discoidea* essential oil. The content of the main phenolics was measured by spectrophotometry. The cytotoxic concentrations of the alcohol extracts and essential oils of *M. discoidea* and *M. chamomilla* flowers showed similar profiles on cell lines, with the essential oils being significantly more toxic than the alcohol extracts. The results of the molecular docking analyses of the identified BASs of *M. discoidea* demonstrated a high probability of COX-1,2 inhibition and NMDA receptor antagonism. The *M. discoidea* dry extracts showed non-toxicity, analgesic activity, and soporific activity. It is evident that these latter two effects are associated with each other. The *M. discoidea* herb dry extract prepared with a 70% ethanol solution had the highest analgesic and soporific activities in rodents.

## Figures and Tables

**Figure 1 biomolecules-14-00361-f001:**
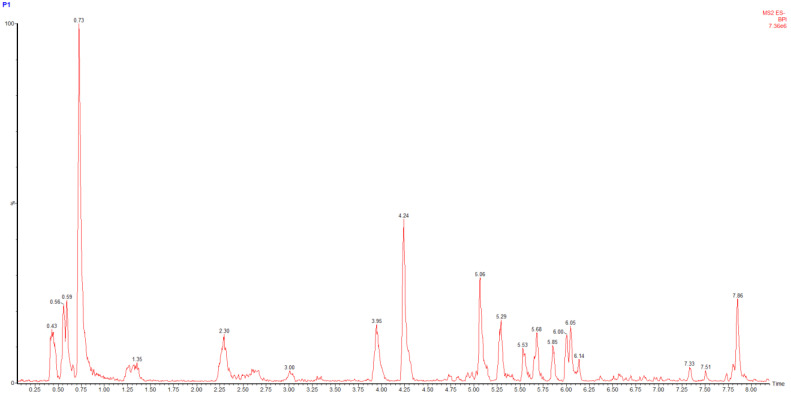
HPLC-MS scan chromatogram in negative ESI mode for the P1 extract as an example.

**Figure 2 biomolecules-14-00361-f002:**
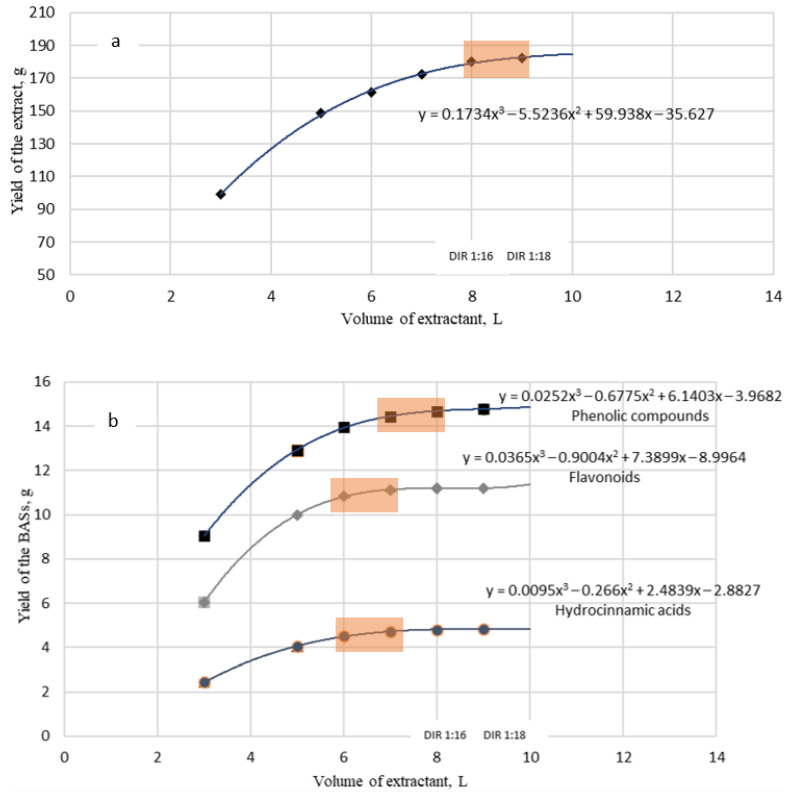
The impact of DIR ratio on the extractive substance yield (**a**) and BAS yield (**b**) from *M. discoidea* during 70% ethanol extraction.

**Figure 3 biomolecules-14-00361-f003:**
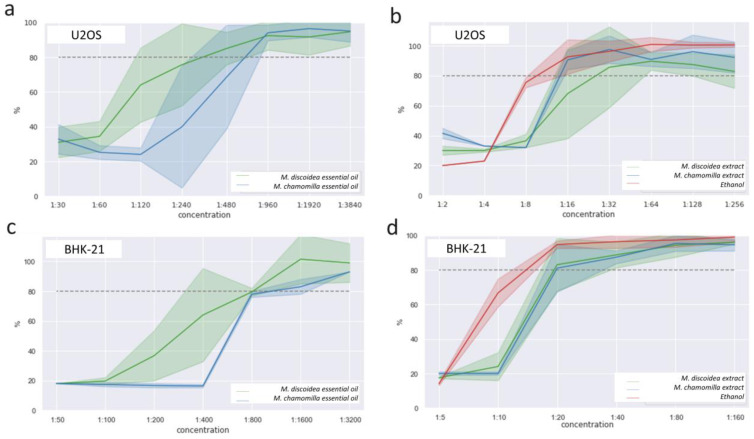
The cytotoxic effects of the essential oils and the extracts of *M. discoidea* and *M. chamomilla* at different concentrations on U2OS (**a**,**b**) and BHK-21 (**c**,**d**) cells after 48 h of incubation. The data represent three biological replicates from four (U2OS) or five (BHK-21) independent experiments, shown as the confidence interval from the z-test. If 80% or more cells survived (dotted line on figure), then the concentration was considered not cytotoxic.

**Figure 4 biomolecules-14-00361-f004:**
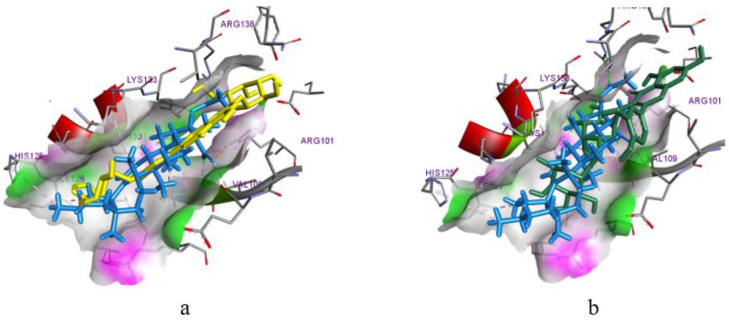
Joint conformational placement of (**a**) luteolin-3,7-diglucoside (yellow molecule), (**b**) rutin (green molecule), and the native inhibitor AKBA (blue molecule) in the active site of LOX-5.

**Figure 5 biomolecules-14-00361-f005:**
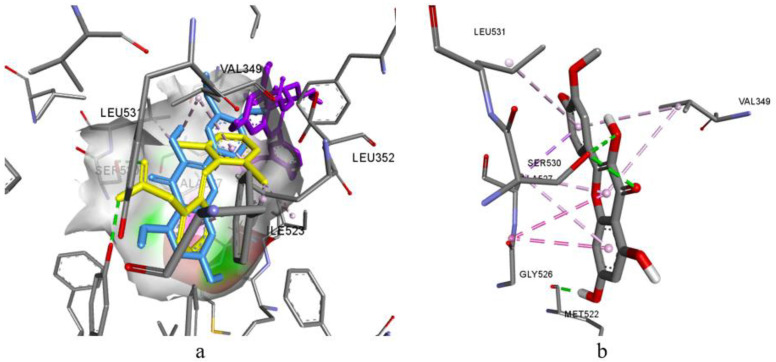
(**a**) Joint conformational placement of isorhamnetin (blue), isorhamnetin-3-glucoside (purple), and diclofenac (yellow) in the active site of COX-1; and (**b**) interaction of isorhamnetin with amino acid residues of COX-1.

**Figure 6 biomolecules-14-00361-f006:**
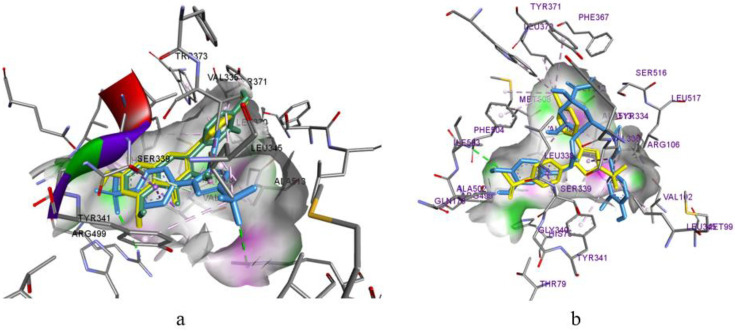
(**a**) Joint conformational placement of isorhamnetin (green), luteolin (blue), and celecoxib (yellow) in the active site of COX-2; and (**b**) interaction of 4,5-dicaffeoylquinic acid with amino acid residues of COX-2.

**Figure 7 biomolecules-14-00361-f007:**
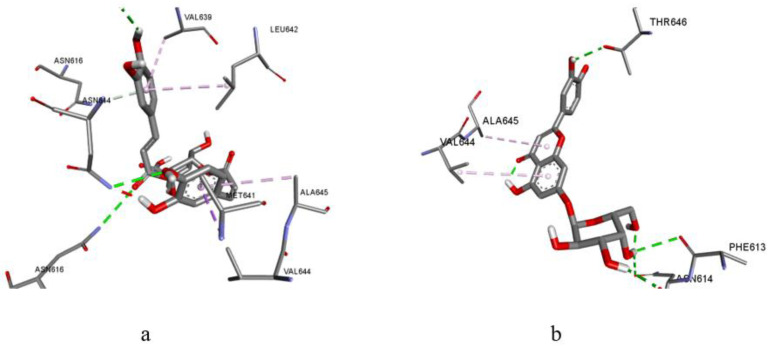
Visualization of the interaction of 4,5-dicaffeoylquinic acid (**a**) and luteolin-7-O-glucoside (**b**) with amino acid residues in the active site of the NMDA receptor.

**Table 1 biomolecules-14-00361-t001:** The content of main terpenoids in the *M. discoidea* essential oil.

RI (DB-5)	Compound	Content in the Oil, %	
		*M. Chamomilla **	*M. Discoidea*
987	Myrcene	<0.01	7.99
1455	(*E*)-ß-Farnesene	24.72	42.51
1472	Germacrene D	1.01	1.23
1570	Spathulenol	2.39	1.12
1609	Geranyl isovalerate	<0.01	29.50
1649	α-Bisabolol oxide B	22.27	1.06
1673	α-Bisabolone oxide A	10.40	2.11
1715	Chamazulene	7.89	-
1740	α-Bisabolol oxide A	21.78	1.48
1874	(*Z*)-Enyne-dicycloether	8.26	8.86
Total		98.72	95.86

* These numbers were obtained from [31].

**Table 2 biomolecules-14-00361-t002:** Content of main phenolics in *M. discoidea* extracts.

Substance	Retention Time, min	Content in the Extract		
		P1	P2	P3
**UPLC-MS/MS, µg/g of dry extract**				
Neochlorogenic acid	2.61	2109.57 ± 70.12	474.21 ± 4.02	805.71± 32.49
Luteolin	7.12	271.53 ± 24.12	1927.41 ± 70.51	114.13 ± 25.62
Cryptochlorogenic acid	3.86	19.81 ± 2.66	228.8 ± 17.58	23.44 ± 3.11
Luteolin-4-O-glucoside	6.05	6.93 ± 1.07	9.27 ± 1.98	0
Chlorogenic acid	3.95	3148.29 ± 143.312	10,836.74 ± 203.23	2202.01 ± 20.64
Isorhamnetin-3-glucoside	5.80	49.65 ± 3.11	40.52 ± 7.19	18.79 ± 1.86
Luteolin-3,7-diglucoside	5.02	117.36 ± 5.927	157.59 ± 2.80	21.69 ± 2.3
Vanillic acid	4.28	23.87 ± 2.87	22.45 ± 1.19	14.25 ± 1.18
Caffeic acid	4.32	37.32 ± 3.81	32.33± 3.26	51.82 ± 5.66
3,4-Dihydroxyphenylacetic acid	2.30	335.69 ± 9.49	117.88 ± 7.33	146.11 ± 7.11
Isorhamnetin	7.95	6.6 ± 0.39	26.96 ± 2.32	8. 4 ± 1.29
Hyperoside	5.42	139.61 ± 1.91	194.14 ± 17.13	51.95 ± 0.93
Luteolin-7-O-glucoside	5.56	2844.8± 212.97	8101.17 ± 1237.03	766.53 ± 188.39
4,5-Dicaffeoylquinic acid	5.68	3339.61 ± 52.33	3049.98 ± 143.4	925.79 ± 48.57
3,5-Dicaffeoylquinic acid	6.06	1708.29 ± 26.77	1578.86 ± 99.56	471.56 ± 26.17
3,4-Dicaffeoylquinic acid	5.85	3502.78 ± 54.88	3233.96 ± 208.24	967.68 ± 54.97
**Spectrophotometry, %**				
Phenolic compounds		5.62 ± 0.06	10.74 ± 0.39	3.17 ± 0.08
Hydrocinnamic acids		1.55 ± 0.28	3.31 ± 0.25	0.98 ± 0.31
Flavonoids		2.37 ± 0.13	8.09 ± 0.54	0.28 ± 0.06

Notes: P1—the dry extract after distillation of essential oil; P2—the dry extract obtained with 70% ethanol solution; P3—the dry extract after obtaining tincture.

**Table 3 biomolecules-14-00361-t003:** Dynamics of phenolic, hydroxycinnamic acid, and flavonoid extraction from *M. discoidea* with 70% ethanol.

Extraction Stage	Dry Residue, %	% in the Dry Residue		
		Phenolic Compounds	Hydrocinnamic Acids	Flavonoids
1	3.57 ± 0.88	12.11 ± 0.22	2.73 ± 0.07	8.75 ± 0.38
2	1.77 ± 0.52	10.74 ± 0.16	3.91 ± 0.13	9.47 ± 0.11
3	1.13 ± 0.09	8.96 ± 0.25	3.73 ± 0.26	5.13 ± 0.15
4	0.67 ± 0.25	7.53 ± 0.53	2.59 ± 0.39	2.88 ± 0.04
5	0.3	7.00 ± 0.15	1.55 ± 0.19	2.25 ± 0.16
6	0.2	4.84 ± 0.18	1.01 ± 0.12	1.29 ± 0.03

**Table 4 biomolecules-14-00361-t004:** Predicted affinity of BASs identified in *M. discoidea* to the active sites of biotargets.

Ligand	Biotargets				
	LOX-5 (6NCF)	COX-1 (3N8Y)	COX-2 (3LN1)	NMDA (7EU7)	ГАМКА (6X3W)
AKBA	−10.0	–	–	–	–
Diclofenac	–	−8.5	-8.4	–	–
Celecoxib	–	–	−12.2	–	–
Ketamine	–	–	–	−5.6	–
Phenobarbital	–	–	–	–	−7.3
Neochlorogenic acid	−7.9	−7.1	−7.5	−7.1	−6.8
Chlorogenic acid	−7.8	−7.1	−7.5	−6.9	−6.8
Cryptochlorogenic acid	−7.8	−6.6	−7.9	−7.0	−6.4
Luteolin	−8.1	−8.1	−9.8	−7.4	−6.6
Luteolin-4-O-glucoside	−9.0	−5.6	−8.6	−8.1	−6.0
Luteolin-7-O-glucoside	−9.6	−5.4	−6.2	−8.4	−6.5
Luteolin-3,7-diglucoside	−9.7	−5.3	−6.8	−7.9	−6.5
Isorhamnetin-3-glucoside	−7.8	−1.8	−8.8	−7.9	−6.5
Vanillic acid	−6.7	−6.2	−6.4	−4.9	−5.1
Caffeic acid	−6.0	−6.5	−7.4	−5.1	−5.0
3,4-Dihydroxyphenylacetic acid	−6.7	−6.1	−6.6	−5.2	−4.9
Isorhamnetin	−7.9	−7.8	−9.6	−7.3	−6.4
Rutin	−8.9	−0.6	−3.7	−9.1	−6.2
Hyperoside	−8.6	−2.1	−8.2	−7.7	−6.6
4,5-Dicaffeoylquinic acid	−8.8	−6.1	−9.1	−8.1	−7.0
3,5-Dicaffeoylquinic acid	−9.0	−6.8	−8.5	−7.9	−7.5
3,4-Dicaffeoylquinic acid	−8.8	−6.0	−8.8	−7.9	−7.3

**Table 5 biomolecules-14-00361-t005:** The interactions between ligands with the best scores and amino acid residues of biotargets.

LOX-5 (6NCF)	Luteolin- 3,7-diglucoside	a: Thr104, His130, Leu163, Glu134, Pro164; b: Thr137, Val107(3); c: Arg101(Pi-Cation).
	Rutin	a: Arg68, Arg101, Glu134, His130, Thr137; b: Lys133, Val107(3).
	3,5-Dicaffeoylquinic acid	a: Arg68, Arg101, Val110, His130, Asp166, Glu108; b: His130, Leu66, Val107.
COX-1	Luteolin	a: Ser530(2), TYR385; b: Ala527(4), Gly526(2), Val349(2), Leu531.
	Isorhamnetin	a: Ser530(2), Met522, Ala527; b: Gly526(2), Ala527(4), Val349(2), Leu531.
	Chlorogenic acid	a: Tyr385, Ser530, Tyr385, Met522; b: Val349, Leu359, Ala527, Leu531.
COX-2	Isorhamnetin	a: Tyr341, Ser516, Ser339, Tyr371; Leu338(2); b: Val509, Val335.
	Luteolin	a: Tyr341, Ser516, Ile503, Phe504, Tyr371; b: Leu338, Val509(2), Leu338, Val335.
	4,5-Dicaffeoylquinic acid	a: Arg106, Tyr371, Gly512 b: Val509(3), Tyr341, Val102, Leu345, Ala502.
NMDA	4,5-Dicaffeoylquinic acid	a: Asn616(2), Asn614, Leu611, Asn616; b: Val644, Val639, Leu642, Met641, Ala645.
	Luteolin	a: Phe613, Leu611(2), Asn615(2); b: Val644(2), Val639, Leu642.

**Table 6 biomolecules-14-00361-t006:** Analgesic activity of the *M. discoidea* extracts in mice (n = 6).

Agent	Group	Dose, mg/kg	The Time of Response(s)/Analgesic Effect (%) in Comparison to (Reference Drug) and [Control]				
			After Administration in				
			30 min	60 min	120 min	180 min	240 min
Control group	1		7.10 ± 0.32	7.00 ± 0.50	7.05 ± 0.28	6.98 ± 0.52	6.40 ± 0.63
Extract P1	2	25	8.85 ± 0.69/ [25%] (−15%)	9.13 ± 0.77/ [30%] * (−12%)	10.67 ± 0.49/ [51%] * (1%)	10.40 ± 0.55/ [49%] * (9%)	9.12 ± 0.51/ [42%] * (9%)
	3	50	10.15 ± 1.49/ [43%] (−3%)	10.30 ± 1.01/ [47%] * (−1%)	12.15 ± 0.39/ [72%] * (15%)	11.07 ± 0,54/ [58%] * (16%)	9.65 ± 0.28/ [51%] * (15%) #
	4	100	10.67 ± 2.79/ [50%] (2%)	12.07 ± 2.40/ [72%] (16%)	11.12 ± 1.27/ [58%] * (5%)	10.57 ± 1.19/ [50%] * (11%)	8.87 ± 1.27/ [39%] (6%)
Extract P2	5	25	10.63 ± 1.01/ [50%] * (2%)	10.42 ± 0.88/ [49%] * (0%)	10.72 ± 0.62/ [52%] * (1%)	10.47 ± 0.67/ [50%] * (10%)	9.48 ± 0.92/ [48%] * (13%)
	6	50	10.98 ± 0.58/ [55%] * (5%)	11.67 ± 0.53/ [67%] * (12%)	12.78 ± 1.87/ [81%] * (21%)	11.72 ± 1.76/ [68%] * (23%)	10.10 ± 1.20/ [58%] * (20%)
	7	100	11.65 ± 1.46/ [64%] (12%)	12.72 ± 1.58/ [82%] * (22%)	12.55 ± 1.53/ [78%] * (19%)	10.30 ± 0.94/ [47%] * (8%)	9.93 ± 1.01/ [55%] * (18%)
Extract P3	8	25	8.97 ± 0.83/ [26%] (−14%)	9.42 ± 1.31/ [35%] (−10%)	9.93 ± 1.11/ [41%] * (−6%)	9.57 ± 0.74/ [37%] * (1%)	9.00 ± 0.79/ [41%] * (7%)
	9	50	7.98 ± 0.47/ [12%] (−24%) #	9.85 ± 1.17/ [41%] (−6%)	10.37 ± 1.21/ [47%] (−2%)	10.08 ± 0.99/ [44%] (6%)	8.12 ± 1.02/ [27%] (−3%)
	10	100	9.07 ± 0.77/ [28%] (−13%)	10.33 ± 0.65/ [48%] * (−1%)	11.03 ± 0.75/ [57%] * (4%)	10.25 ± 1.10/ [47%] * (8%)	8.75 ± 0.60/ [37%] * (4%)
Acetaminophen	11	50	10.45 ± 0.73 [45%] *	10.43 ± 0.59 [49%] *	10.57 ± 0.71 [50%] *	9.50 ± 0.57 [36%] *	8.38 ± 0.33 [31%] *

* Statistically significant (*p* < 0.05) compared to the control group using Student’s criterion. # Statistically significant (*p* < 0.05) compared to the group receiving 50 mg/kg of acetaminophen using Student’s criterion.

**Table 7 biomolecules-14-00361-t007:** The impact of *M. discoidea* extracts on the period of a sleep induced by sodium thiopental, t ± Δt (n = 6).

Agent	Group	Dose, mg/kg	Average Sleep Duration, min	Soporific Activity, %
Control group	1	40	87.33 ± 11.56	100.0%
Extract P1	2	25	180.17 ± 11.37 *	206.3%
	3	50	171.67 ± 2.87 *	196.6%
	4	100	170.00 ± 9.27 *	195.8%
Extract P2	5	25	243.00 ± 8.07 *#	278.2%
	6	50	215.50 ± 10.57 *#	246.8%
	7	100	248.67 ± 6.10 *#	284.7%
Extract P3	8	25	165.67 ± 12.26 *	189.7%
	9	50	156.17 ± 10.81 *#	178.8%
	10	100	167.67 ± 10.11 *	192.0%
Valerian extract	11	2.15	185.33 ± 5.42 *	212.2%

* Statistically significant (*p* < 0.05) compared to the group administrated sodium thiopental according to Student’s test. #Statistically significant (*p* < 0.05) compared to the group given “Valerian syrup AN NATUREL” according to Student’s test.

## Data Availability

The raw data supporting the conclusions of this article will be made available by the authors on request.

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
