# Peer review of "Phytochemical, Pharmacological, and Molecular Docking Study of Dry Extracts of Matricaria discoidea DC. with Analgesic and Soporific Activities"

_biomolecules, 2024, doi:10.3390/biom14030361_

Round 1

Reviewer 1 Report

Comments and Suggestions for Authors

A phytochemical, analgesic and soporific activity study of ethanol extracts and essential oil of Matricaria discoidea DC is presented. In addition, molecular docking studies of some secondary metabolites of the extracts were carried out, analyzing the COX-1,2 inhibition capacity and the possible antagonistic effects of the dNMDA receptor.

The document is well developed, the title is clear and forceful, the summary contextualizes the reader, clearly describes the objective of the research, reports the methodological basis, the most important results and some relevant conclusions.  

As for the introduction, in addition to broadening the contextualization of the subject, it clearly describes the knowledge gap and presents a good justification between the lines. 

The methodological section is well developed, the step-by-step of each of the procedures is presented, and the bibliographic bases of these procedures are reported.  This guarantees the quality of the results and therefore the analysis of results and conclusions. 

The results are generally well described and analyzed, demonstrating an important interdisciplinary work, since the researchers perform an extraction, chemically analyze each extract, develop a pharmacological study to evaluate analgesic and soporific properties, and culminate with molecular docking studies. Each of the studies is well correlated and provides a complete picture of the research. In relation to the ethical regulations, the guidelines and approval of an ethics committee are established as required for the development of this type of studies. 

Finally, I have some minor qualms, which I will describe below.

1. Write the entire document in the third person.

2. It would be important to report the characteristics and/or commercial information of all reagents used. 

3. In relation to the pharmacopoeia of the USSR... I do not know to what extent it is relevant, however it is important to make it clear that it is a historical concept (I suppose that currently this pharmacopoeia is not in force). 

4. It would be important to report some chromatograms of the HPLC and Mass-Gas studies.

5. Figure two should be improved, the resolution is very poor. 

Author Response

Manuscript ID: biomolecules-2886758
The dry extracts of Matricaria discoidea DC. herb with analgesic and soporific activity: phytochemical, pharmacological and molecular docking research

Reviewer 1

The authors of the manuscript appreciate your thorough feedback. We have updated and improved the manuscript according to your helpful comments. The changes made in the manuscript are highlighted in yellow colour. We also provide our responses to your comments.

Comment 1: Write the entire document in the third person.
Response 1: We thank you for this comment. The manuscript has been corrected to third person.

Comment 2: It would be important to report the characteristics and/or commercial information of all reagents used.
Response 2: We improved the manuscript by adding relevant information about all reagents used.

Comment 3: In relation to the pharmacopoeia of the USSR... I do not know to what extent it is relevant, however it is important to make it clear that it is a historical concept (I suppose that currently this pharmacopoeia is not in force).
Response 3: Thank you. The status of the USSR pharmacopoeia was clarified and the sentences rephrased (L47-50). This is just a historical point of view.

Comment 4: It would be important to report some chromatograms of the HPLC and Mass-Gas studies.
Response 4: An example of HPLC-MS scan chromatogram (Figure 1) was added to manuscript (L433). Table 2 was improved by adding retention time as a relevant parameter (L435). Unfortunately, the quality of GC-chromatogram is so poor that we can not add it.

Comment 5: Figure two should be improved, the resolution is very poor.
Response 5: The figure was improved to a better quality (Figure 3. L502), it have now the resolution more than 300 dpi..

Reviewer 2 Report

Comments and Suggestions for Authors

The authors of this manuscript characterized the analgesic and soporific activity, as well as the phytochemical, pharmacological, and molecular docking studies of the dry extracts and essential oil of Matricaria discoidea.

The manuscript was well-designed and added new information about the monograph for the herbal drug and oil of M. discoidea. The study is also important in the utilization of this herb in the pharmaceutical or food industries. However, there are some points that need to be clarified and corrected.

Comments

1. L44: Asteraceae should not be italic.

2. L61: M. discoidea and M. chamomilla should be italic.

3. L111: Please identify which plant part of M. chamomilla and M. discoidea you used in essential oil extraction.

4. L167: 30 ml/min flow rate. This needs to be checked.

5. The methods used to assay the total phenols, hydroxycinnamic acids, and flavonoids should be written in detail.

6. L186: was conducted

7. L191:  Please write the gradient elution based on solvent B, as it is the organic solvent (acetonitrile).

8. L196: were a capillary voltage

9. L351: delete and

10. L362: extract

11. L373: delete an

12. The quality of Figure 2 needs to be enhanced.

13. L476: an extended sleep effect

14. L484: This is obviously due to the fact that

15. L489: (E)-ß-farnesene

16. Use one name only (Z)-enyne-dicycloether or (Z)-enyne-bicycloether

17. L491: studies showed

18. L498: Please identify the different aerial parts of M. discoidea to correspond with the percentages mentioned in line 499.

19. L502: delete [1] because of repetition.

20. L524: the extractant,

21. L618: The consumption of extract

22. The reference number 51 is not found in the text.

23. An inappropriate self-citations by the co-author (Ain Raal) were detected (about 12 references).

Comments on the Quality of English Language

/

Author Response

Manuscript ID: biomolecules-2886758
The dry extracts of Matricaria discoidea DC. herb with analgesic and soporific activity: phytochemical, pharmacological and molecular docking research

Reviewer 2

The authors of the manuscript appreciate your thorough feedback. We have updated and improved the manuscript according to your helpful comments. The changes made in the manuscript are highlighted in yellow colour. We also provide our responses to your comments.

Comment 1: L44: Asteraceae should not be italic.
Response 1: Thank you very much for this comment. Manuscript corrected (L44).

Comment 2: L61: M. discoidea and M. chamomilla should be italic. Response 2: Thanks again. Manuscript corrected (L62).

Comment 3: L111: Please identify which plant part of M. chamomilla and M. discoidea you used in essential oil extraction.
Response 3: This relevant information was added to manuscript (L120).

Comment 4: L167: 30 ml/min flow rate. This needs to be checked. Response 4: The flow rate was checked and corrected to 3.0 ml/min (L175).

Comment 5: The methods used to assay the total phenols, hydroxycinnamic acids, and flavonoids should be written in detail.
Response 5: The manuscript was improved with a more detailed description of the methods used.

Comment 6: L186: was conducted Response 6: Manuscript corrected (L258).

Comment 7: L191: Please write the gradient elution based on solvent B, as it is the organic solvent (acetonitrile).
Response 7: Relevant information was added to manuscript (L261-266).

Comment 8: L196: were a capillary voltage Response 8: Manuscript corrected (L268).

Comment 9: L351: delete and
Response 9: Manuscript corrected (L423).

Comment 10: L362: extract
Response 10: Manuscript corrected (L437).

Comment 11: L373: delete an
Response 11: Manuscript corrected (L447).

Comment 12: The quality of Figure 2 needs to be enhanced.
Response 12: The figure was improved to a better quality (Figure 3. L502).

Comment 13: L476: an extended sleep effect Response 13: Manuscript corrected (L537).

Comment 14: L484: This is obviously due to the fact that Response 14: Manuscript corrected (L544).

Comment 15: L489: (E)-ß-farnesene Response 15: Manuscript corrected (L549).

Comment 16: Use one name only (Z)-enyne-dicycloether or (Z)-enyne-bicycloether Response 16: Manuscript corrected.

Comment 17: L491: studies showed Response 17: Manuscript corrected (L553).

Comment 18: L498: Please identify the different aerial parts of M. discoidea to correspond with the percentages mentioned in line 499.
Response 18: Relevant information added to the manuscript (L562).

Comment 19: L502: delete [1] because of repetition. Response 19: Repetition deleted.

Comment 20: L524: the extractant, Response 20: Manuscript corrected (L583).

Comment 21: L618: The consumption of extract Response 21: Manuscript corrected (L677).

Comment 22: The reference number 51 is not found in the text.

References 51 and 4 are the same

Response 22: The references were checked and corrected.

ones. So, 51 was removed. The order was changed.

Comment 23: An inappropriate self-citations by the co-author (Ain Raal) were detected (about 12 references).
Response 23: The number of self-citations was reduced. From 12 to 9.

Round 2

Reviewer 2 Report

Comments and Suggestions for Authors

All comments were addressed by the authors, and the manuscript was highly improved.

Thank you.